# Chloroplast Genome-Based Hypervariable Markers for Rapid Authentication of Six Korean Pyropia Species

**Sung-Je Choi [1,†], Yonguk Kim [2,†] and Chulyung Choi [2,*]**

[1] Ocean and Fisheries Science Institute Haenam Branch, Haenam-gun, Jeollanamdo 59046, Korea; csjchoi@korea.kr

[2] Jeonnam Institute of Natural Resources Research, Jangheung-gun, Jeollanamdo 59338, Korea; kyu9801@hanmail.net

[†] These authors contributed equally to this work.

[*] Correspondence: blockstar@hanmail.net; Tel.: +82-61-860-2620

**Abstract:** We previously established that polymerase chain reaction-restriction fragment length polymorphism (PCR-RFLP) analysis using partial plastid *rbcL* and mitochondrial *trnC–trnP* gene sequences can be used to distinguish the six representative *Pyropia* species produced via mariculture in Korea. In this study, we develop progressive InDel markers by comparing seven complete *Pyropia* chloroplast genomes obtained from The National Center of Biotechnology Informnation (NCBI) GenBank. Comparative analyses of nucleotide diversity among the genomes revealed seven hypervariable sites (*cemA, rps13, trnM-argB, petD-petB, trnR-trnQ, ccs1-orf24,* and *ycf12-ftrB*) among 637 sliding windows with nucleotide diversity > 0.025 (Pi). These sites included two genes and five gene-intergenic regions, three of which (*cemA, trnM-argB, trnR-trnQ*) showed complete amplification for all six test species. Finally, *trnM-argB*, an InDel-variable locus with high discriminatory power, was selected as a DNA barcode candidate. These results suggest that the obtained *trnM-argB* region can be used for the effective exploration of the variation present in six Korean *Pyropia* and for further evolutionary, phylogenetic, barcoding and genetic engineering studies of *Pyropia* species.

**Keywords:** *Pyropia*; chloroplast genome; InDel; nucleotide diversity; DNA barcode

## 1. Introduction

*Pyropia*, a genus of red algae (Bangiales, Rhodophyta), is an economically important mariculture crop and food source with a long history of consumption in China, Japan, and South Korea [1]. This genus is widely used as a laver or dried sheet product, "zicai" in China, "nori" in Japan, and "gim" in South Korea [2].

Among twelve species of *Pyropia* distributed in South Korea, *Pyropia yezoensis*, *Pyropia seriata*, *Pyropia dentata*, and *Pyropia suborbiculata* are extensively cultured in the Korean southwest sea, whereas *Pyropia pseudolinearis* is distributed across the Korean East Sea [3]. *Pyropia haitanensis* is one of the Chinese indigenous laver species and has been utilized as a potential genetic source for breeding to develop improved cultivars that are optimized for the warm temperature zones in Korean mariculture industries.

In Korean aquaculture industry, *Pyropia* species are identified using morphological and anatomical features. Morphological characteristics remain an important tool for phylogenetic studies, even in the current age of molecular systematics; however, the use of classical morphological features

to distinguish *Pyropia* species has some limitations. In addition, for morphologically similar *Pyropia* species such as *P. dentata* and *P. haitanensis* species-specific identification and discrimination in the field is difficult and require a using a common DNA barcoding method.

Thus, in our previous study, we established a PCR-restriction fragment length polymorphism (RFLP) method for the rapid and accurate identification of six representative Korean *Pyropia* species [4]. On the basis of that study, a high level of expertise is required to correctly identify species with the accuracy required in the aquaculture field and marine industry.

In the present study, we identified hypervariable loci by comparing the chloroplast genomes of seven *Pyropia* species, which enabled us to develop valuable chloroplast-based InDel markers for authenticating six Korean *Pyropia* species.

## 2. Materials and Methods

### 2.1. Chloroplast Genome Comparison and Identification of Hypervariable InDels

We downloaded from GenBank all of the chloroplast genome sequences in the *Pyropia* genus with complete genome sequence information (Table 1). The sequences were first aligned using the Clustal W algorithm of MEGA7.0 [5]. The chloroplast genome gene distribution and similarities of *Pyropia* species were compared and visualized using mVISTA software in Shuffle-LAGAN mode with the annotation of *P. haitanensis* NC021189 as a reference [6]. The variability of the aligned genomes was evaluated using the sliding window method in DNAsp ver. 5.0 [7]. The window length was set to 600 bp, the typical length of DNA markers. The step size was set to 250 for relatively accurate positioning of hyper-variable InDels. We only considered regions with nucleotide diversity (*Pi*) of a value > 0.2. Hypervariable sites and genetic distances among the chloroplast genomes were calculated using MEGA 7.0. Based on the aligned sequence matrix, the InDel events were checked manually.

**Table 1.** Complete and partial chloroplast genomes of the seven *Pyropia* species used in this study.

| No. | Species | Research Group | NCBI Accessions of Chloroplast Genome | Sequence Length (bp) |
|-----|---------|----------------|---------------------------------------|----------------------|
| 1 | *P. yezoensis* | National Research Institute of Fishery Science, Aquatic genomics research center, Kanagawa, Japan | NC007932 | 191,952 |
| 2 | *P. haitanensis* | National Center for Biotechnology Information, NIH, USA | NC021189 | 195,597 |
| 3 | *P. endiviifolia* | College of Marine Life Sciences, Ocean University, China | KT716756 | 195,784 |
| 4 | *P. perforata* | | KC904971 | 189,789 |
| 5 | *P. kanakaensis* | Math and Sciences, Hartnell College, Central Ave., USA | KJ776836 | 189,931 |
| 6 | *P. fucicola* | | KJ776837 | 187,282 |
| 7 | *P. pulchra* | Biological Sciences, Sungkyunkwan,University, Korea | KT266789 | 194,175 |

### 2.2. Sample Collection and DNA Isolation

All samples were identified based on our previous study [4] (Table 2). Prior to DNA extraction, conchocelis-stage wet samples of all strains (~200 mg wet weight) were washed twice with distilled water and centrifuged for 20 min at 3000 x g twice. The remaining wet phase was dried between sheets of filter papers at room temperature, and 50 mg dry weight of each sample was added to steal bead tubes (2.38 mm diameter) from the PowerPlant Pro DNA isolation Kit (Qiagen, Valencia, Spain), and then the mixtures were subjected to bead beating using a Precellys Evolution homogenizer (Bertin Technologies, Paris, France). Genomic DNA was extracted from ground conchocelis-stage

samples using the PowerPlant Pro DNA isolation Kit (Qiagen, Valencia, Spain) according to the manufacturer's instructions.

**Table 2.** *Pyropia* samples used in this study.

| No. | Scientific Name (n = 3) | Common Name | Collection Site | Location |
|-----|-----|-----|-----|-----|
| 1 | *P. yezoensis* | Bangsamunuigim | Songji-myeon, Haenam-gun, Jeollanam-do | 34°21′05.92″ N 126°27′40.76″ E |
| 2 | *P. dentata* | Itbadidolgim | Yuldo-dong, Mokpo-si, Jeollanam-do | 34°48′13.22″ N 126°18′34.88″ E |
| 3 | *P. seriata* | Momunuidolgim | Songji-myeon, Haenam-gun, Jeollanam-do | 34°45′49.37″ N 126°07′50.54″ E |
| 4 | *P. suborbiculata* | Dunggeundolgim | Nam-myeon, Yeosu-si, Jeollanam-do | 34°25′32.73″ N 127°47′31.33″ E |
| 5 | *P. pseudolinearis* | Ginipdolgim | Ulleung-gun, Gyeongsangbuk-do | 37°27′31.55″ N 130°54′14.98″ E |
| 6 | *P. haitanensis* | Haitanensisgim | Dried laver product from China | |

*2.3. Development and Validation of the InDel Molecular Marker*

In order to validate interspecies polymorphisms within the chloroplast genomes and develop DNA genetic markers for identifying the six *Pyropia* species studied here, primers were designed using Primer 3 and NCBI Primer-BLAST based on the mutational hotspot regions (hypervariable regions) found in the *Pyropia* chloroplast genomes. PCR amplifications were performed in a reaction volume of 50 μL containing 25 μL premix (Ex Taq Version 2.0, TaKaRa, Japan), 10 ng genomic DNA template, and 1 μL (10 pM) forward and reverse primers. The mixtures were denatured at 95 °C for 5 min and amplified with 40 cycles of 95 °C for 30 s, 55 °C for 20 s, and 72 °C for 30 s, with a final extension at 72 °C for 5 min. For the detection of PCR amplicons, PCR products were separated by capillary electrophoresis (QIAxcel, Qiagen, Valencia, Spain) using the high-resolution kit and the 0M500 method (Qiagen, Valencia, Spain). The target DNA was extracted and purified using a MinElute PCR Purification Kit (Qiagen, Valencia, Spain). Purified PCR products were then sent to CosmoGenetech for sequencing (Seoul, Korea) with both forward and reverse primers. Sequencing results were analyzed by a BLAST search of the GenBank database. Alignment and graphical representation was carried out in CLC Sequence Viewer 8.0.

**3. Results**

The divergent regions of seven *Pyropia* chloroplast genomes were examined with the aim of distinguishing six common *Pyropia* species. First, interspecific comparisons of sequence identity among the seven chloroplast genomes were conducted with mVISTA using the annotated *P. haitanensis* sequence as a reference (Figure 1).

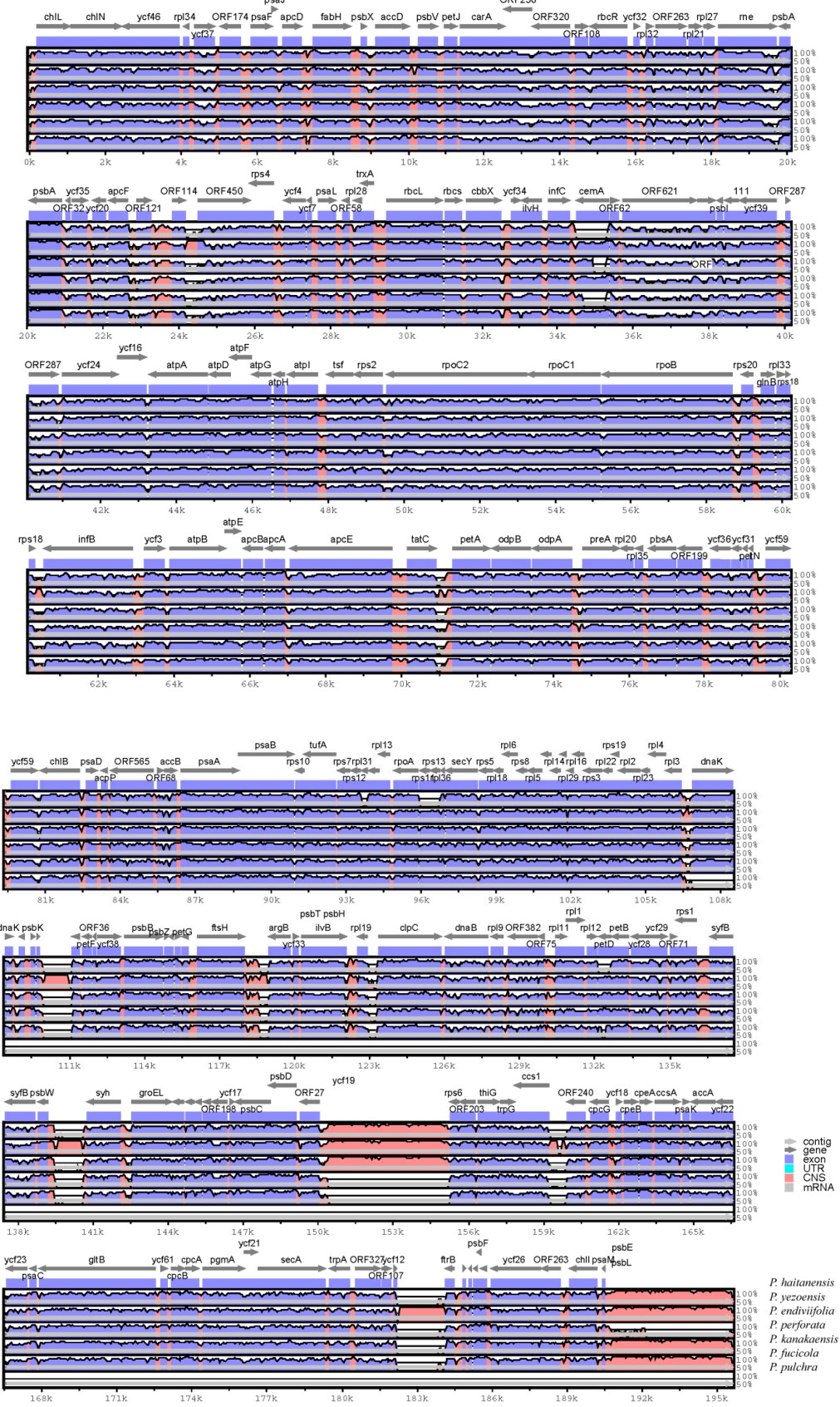

**Figure 1.** Comparison of seven *Pyropia* chloroplast genomes using mVISTA. The complete chloroplast genomes of seven *Pyropia* species obtained from GenBank were compared using *P. haitanensis* as a reference. Blue block: conserved genes; red block: conserved non-coding sequences (CNS). White areas represent regions with sequence variation among the six *Pyropia* species.

The mVISTA results showed that the seven chloroplast genomes were highly conserved; however, the coding gene regions appeared to be more variable than the non-coding sequence (CNS) regions. The overall sequence divergence estimated based on the p-distance among the seven genomes was only 0.1335. The pairwise *p*-distances between the species ranged from 0.1205 to 0.1416. The highest *p*-distance value was due to insufficient regions analyzed between the partial genome sequences of two species, *P. fucicola* KJ776837 and *P. kanakaensis* KJ776836. Furthermore, sliding window analysis using DnaSP detected highly variable regions among the *Pyropia* chloroplast genomes (Figure 2).

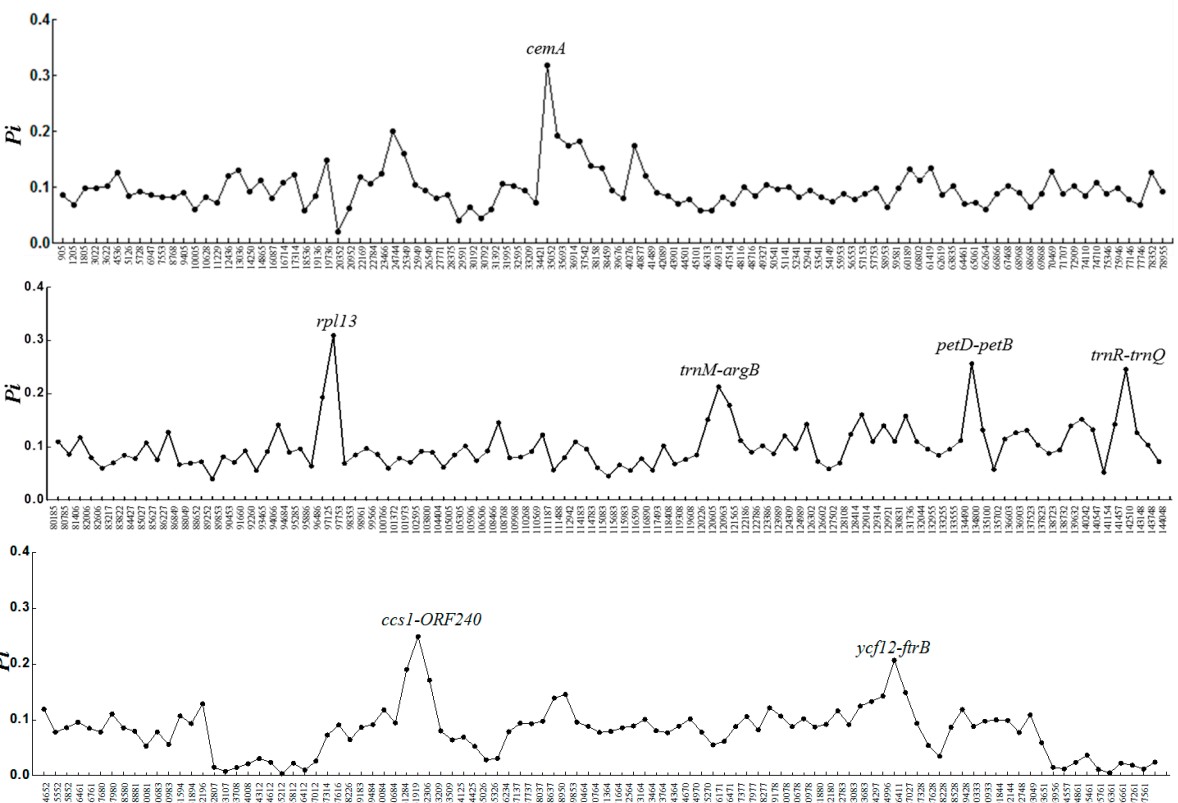

**Figure 2.** Sliding window analysis of the whole chloroplast genomes of seven *Pyropia* species. Window length: 600 bp; step size: 250 bp. X-axis: position of the midpoint of a window; Y-axis: nucleotide diversity of each window (*Pi*).

The average value of nucleotide diversity (*Pi*) was 0.09601. Among the 637 windows, there were seven mutational hotspots that showed remarkably higher *Pi* values (>0.2), including two gene regions (*cemA*, 0.31105; *rps13*, 0.31989) and five intergenic regions (*trnM-argB*, 0.21322; *petD-petB*, 0.24061; *trnR-trnQ*, 0.24611; *ccs1-ORF240*, 0.24944; *ycf12-chlI*, 0.20689) (Figure 2).

Among these seven hypervariable regions, we identified InDel regions by PCR amplification and sequencing to discriminate between six of the *Pyropia* species (Table 3).

The specific primer pairs for three loci (*cemA*, *trnM-argB*, *trnR-trnQ*) successfully amplified their target in all six species. For the other loci, we failed to amplify sequences as follows: *rpl13* of *P. yezoensis*; *petD-petB* of *P. yezoensis*, *P. haitanensis*, and *P. pseudolinearis*; *ccs1-ORF240* of *P. yezoensis* and *P. pseudolinearis*; *ycf12-ftrB* of *P. yezoensis*, *P. dentata*, *P. haitanensis*, and *P. seriata* (Figure 3).

**Table 3.** Primers for amplifying and sequencing seven highly variable loci.

| No. | Locus | Location | Forward Primer (Sequence 5′ to 3′) | Reverse Primer (Sequence 5′ to 3′) | Product Range (bp) | AS | No. of InDels | Mean Pairwise Distance |
|-----|-------|----------|------------------------------------|------------------------------------|--------------------|-----|---------------|------------------------|
| 1 | *cemA* | 35053..35693 | ATTGCAATTTGNCTTTGTCCAG | GAAAAAGTTGGGCCAATACCTA | 506–529 | 100 | 4 | 0.108 |
| 2 | *rpl13* | 96807..97443 | AACACCNTTAACTGCATTACGTT | GTCACNGAAAAGTCATGGTAATT | 583–600 | 83.3 | N/A | N/A |
| 3 | *trnM-argB* | 120227..120963 | TGAGCTACTGAGCCATAATA | CTGATCAAGGTATTGGCTCGAT | 341–696 | 100 | 51 | 0.295 |
| 4 | *petD-petB* | 134166..134800 | CTTCTAAAAGGATTTTGAAACTT | CAGATGCTGTTCCAGTTGTTGGA | 629 | 50 | N/A | N/A |
| 5 | *trnR-trnQ* | 141451..143365 | GGTTGTAGCTCAGANGGATAG | GGGTGTAGCCAAGTGGTAAG | 534–1260 | 100 | 37 | 0.572 |
| 6 | *ccs1-orf24* | 161285..162306 | TGTTCAATAATAGTTCCTATAATGC | TGGAATAATCTNTGGGCTCCTTT | 870–916 | 66.6 | N/A | N/A |
| 7 | *ycf12-ftrB* | 184298..186411 | GAAAAGAGGCAATCTTTAGTAAT | TGGAACTGNCCATGTGTACCAATG | 583 | 16.6 | N/A | N/A |

AS: PCR amplification success (%); mean pairwise distance*: p*-value of each locus based on multiple sequence alignment of six *Pyropia* species.

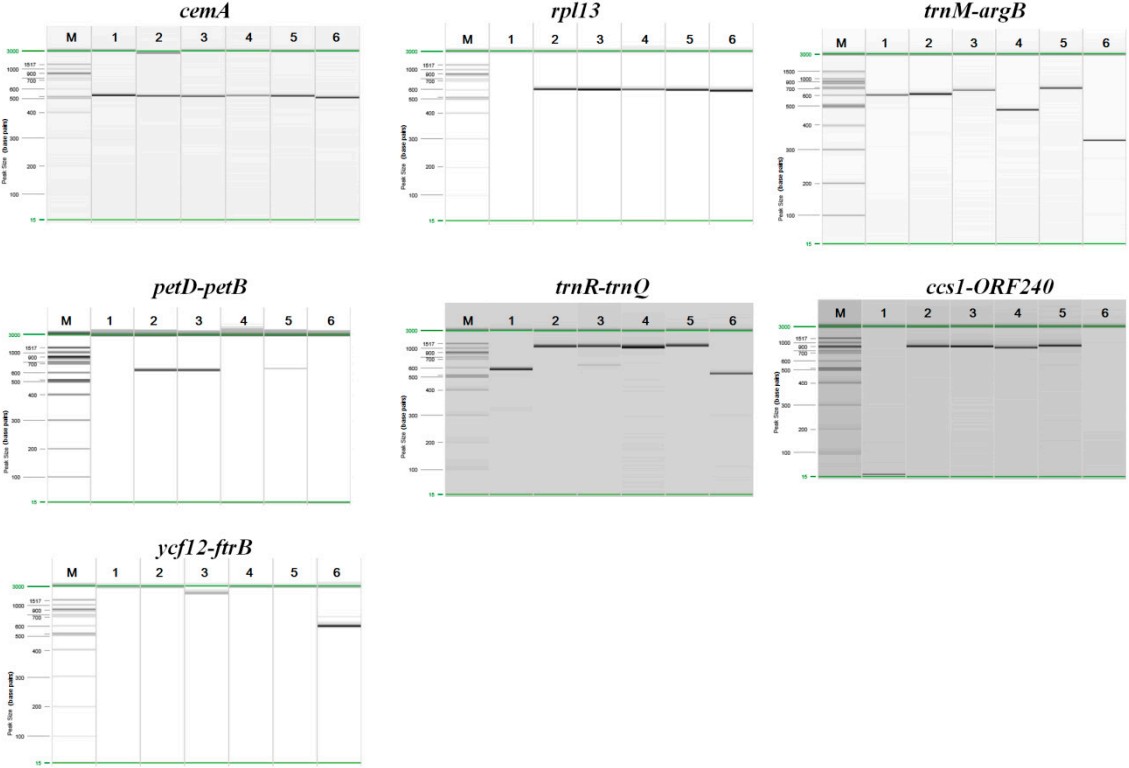

**Figure 3.** Gel profiles of fragments amplified from six species using seven pairs of primers derived from hypervariable regions of seven *Pyropia* chloroplast genomes. M indicates the lane containing the molecular weight marker. The numbers correspond to the species listed in Table 2.

The *cemA* primers resulted in PCR products of 529, 522, 520, 527, 523, and 506 bp from *P. yezoensis*, *P. dentata*, *P. suborbiculata*, *P. haitanensis*, *P. seriata*, and *P. pseudolinearis*, respectively, and the *trnM-argB* primers resulted in products of 597, 629, 655, 484, 696, and 341 bp, respectively. The product sizes for the *trnR-trnQ* primers were as follows: 579 bp from *P. yezoensis*; 1166 bp from *P. dentata*; 1198 bp from *P. suborbiculata*; 1061 bp from *P. haitanensis*; 1260 bp from *P. seriata*; and 534 bp from *P. Pseudolinearis* (Table 3 and Figure 3).

We considered 600 bp to be the acceptable length of the sequence read of a given PCR product. To evaluate the sequence divergence between six *Pyropia* species at the three hypervariable InDel regions, the average pairwise distance values were calculated for each marker using MEGA7.0. The *trnR-trnQ* locus was the most divergent with a maximum pairwise distance of 0.572, followed by *trnM-argB* (0.295). The lowest genetic distance was observed at *cemA* (0.108), indicating that there are no differences between the six species at that locus. Therefore, the species-specific primers derived from the *trnR-trnQ* and *trnM-argB* regions are considered useful markers for elucidating the phylogenetic relationships of the six *Pyropia* species analyzed. However, when selecting attractive InDel markers, the length of the amplified regions must also be considered. The length of only one region, *trnM-argB*, is considered relatively short and sufficient to reproduce the nucleotide variation in *Pyropia* taxa (Figure 4).

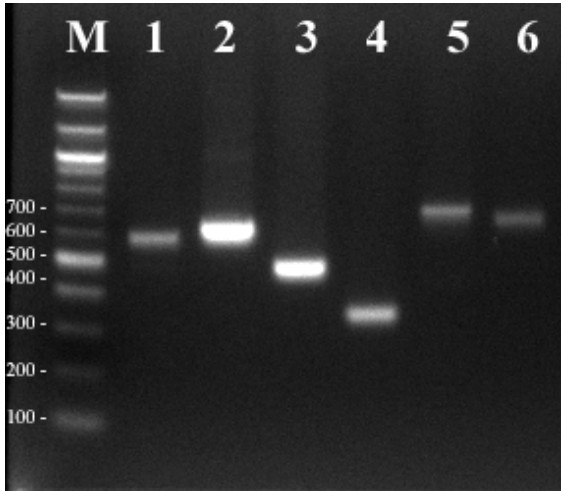

**Figure 4.** PCR verification of *trnM-argB* InDel markers for six *Pyropia* species. M, 1 kb DNA ladder; 1, *P. yezoensis*; 2, *P. dentata*; 3, *P. suborbiculata*; 4, *P. haitanensis*; 5, *P. seriata*; 6, *P. pseudolinearis*.

## 4. Discussion

In our previous work, we found that six restriction enzymes were predicted to show species-specific RFLP patterns and could be used to identify the six *Pyropia* species using *rbcL* in *P. yezoensis* and *rps11-sdh3* in *P. seriata*, *P. pseudolinearis*, *P. dentata*, *P. suborbiculata*, and *P. haitanensis*[4]. This work demonstrated that the PCR-RFLP method is an efficient tool for discriminating between these six Korean *Pyropia* species, and that it avoids confusion caused by surface contamination from symbiotic and invasive species. However, the RFLP method has limited target sensitivity and is time consuming, due to the researcher-dependent nature of the method. Nowadays, the PCR-RFLP method is being replaced by more practical and faster methods utilizing chloroplast genome-based markers.

In recent years, mitochondrial *CO1* and plastid *rbcL* have become the most commonly used genes for red algal barcoding [8,9]. However, the resolution of these markers may not be optimal in some lineages, particularly at the population level, with *rbcL* known to provide lower resolution than *CO1* in some cases [10,11]. A comprehensive comparison and evaluation of new species-specific markers and their potential ability to discriminate species, subspecies, and populations is significant, but few have been assessed [12]. Many researchers began by comprehensively investigating new or combined species-specific variable markers in the chloroplast region of *Pyropia*. Xu et al. 2018 utilized a set of chloroplast whole genomes to explore the phylogenetic relationships of 10 *Pyropia* species [13]. Research has increased the availability of genetic resources for *Pyropia*, such as useful molecular markers or DNA barcodes for species identification using the chloroplast genomes. However, systematic molecular investigations and genomic information on *Pyropia* remain incomplete compared to those of many land plants. In public databases such as NCBI GenBank, only seven complete or partial chloroplast (plastid) genomes within the *Pyropia* genus have been reported for *Pyropia* species, including *P. yezoensis*, *P. haitanensis*, *P. fucicola*, *P. pulchra*, *P. kanakaensis*, *P. perforata*, and *P. endiviifolia*. Information on the chloroplast genomes of Korean red algae, including representative *P. dentata* and *P. seriata*, is very limited.

Using seven previously identified *Pyropia* chloroplast genomes to assess the genetic nucleotide diversity between six *Pyropia* species, we identified and validated three hypervariable regions, from which we developed species-specific InDels in the *trnM-argB* region. Although the present study analyzed a limited number of *Pyropia* species and chloroplast genomes, further studies should include various other species and chloroplast genomes to further elucidate the phylogenetic relationships of the *Pyropia* genus. The markers developed here can now be employed to explore the variations of *Pyropia* populations for further evolutionary, phylogenetic, and barcoding studies in marine fields.

**Author Contributions:** conceptualization, S.-J.C., Y.K., and C.C.; methodology, Y.K.; investigation, S.-J.C. and Y.K.; resources, S.-J.C.; writing-original draft preparation, Y.K. and S.-J.C.; writing-review and editing, Y.K.

**Funding:** This research received no external funding.

**Acknowledgments:** This paper is dedicated to the memory of Heungsu Kim for inspiring and mentoring our work and teaching us as a great inventor and scientist with a passion for plant biology.

**Conflicts of Interest:** The authors declare no conflict of interest.

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
