# Peer review of "Chloroplast Genome-Based Hypervariable Markers for Rapid Authentication of Six Korean Pyropia Species"

_diversity, doi:10.3390/d11120220_

Round 1

Author Response

Response to Reviewer#1:

We, the authors of diversity-639877, would like to thank the reviewer for careful and thorough reading of this manuscript and for the thoughtful comments and constructive suggestions, which help to improve the quality of this manuscript. We believe that the comments and the clarity of our paper are much improved in the revised version. Our response follows (the reviewer's comments are in italics).

General Comments. (1) only one primer set appears to have been developed for each identified marker; (2) primer pair prediction does not appear to have been based on alignments therefore affecting the efficiency of their PCR assays, e.g., ambiguous primers could be developed for higher PCR assay success (however, this would not be an issue if the regions surrounding the markers have a 100% identity); (3) the primers were tested on a limited number of samples, only six; and (4) the actual test samples come from species that are different from those used to develop the primers (those whose plastid genomes are sequenced); testing these primers on samples that come from those whose genomes are sequenced could serve as a good reference and control.

Reply:

We appreciate the positive feedback from the reviewer.
(1),(2): As this results, among seven primer sets, we chose only one primer set (trnM-argB) for detecting and identifying six Korean Pyropia species using by PCR method in the aquaculture industry. 

(3): we tested three individuals in six Pyropia species. 

(4): I would greatly appreciate your opinion and suggestion.

General remark.
Please include images of the actual gels made in the Results. I consider this a more important request.

Reply:

As suggested by the reviewer, we have added Figure 4; PCR verification of trnM-argB InDel markers for six Pyropia species.

Minor comments:

1) It would be useful if the authors include more background information on what previous efforts have been made to generate molecular markers for the discrimination of different Pyropia species.

Reply:

Discussion section exist to answer the reviewer's suggestion;

"In our previous work, we found that six restriction enzymes were predicted to show species-specific RFLP patterns and could be used to identify the six Pyropia species using rbcL in P. yezoensis and rps11-sdh3 in P. seriata, P. pseudolinearis, P. dentata, P. suborbiculata, and P. haitanensis. This work demonstrated that the PCR-RFLP method is an efficient tool for discriminating between these six Korean Pyropia species and avoids confusion caused by surface contamination from symbiotic and invasive species."

2) Fix overlapping gene names in Fig. 1.

Reply:

The suggested correction has been made.

3) L102-103: cemA and ‘rpl13’ are the only coding regions that are identified as being variable. All other regions correspond to non-coding intergenic regions.

Reply:

The suggested correction has been made.

4) Fig. 2: Correct label to ccs1-ORF240. This is according to the genome annotation in Fig.1.

Reply:

The suggested correction has been made.

Reviewer 2 Report

The manuscript “Chloroplast genome-based hypervariable markers for rapid authentication of six Korean Pyropia species” by Choi et al. presents several chloroplast ‘hypervariable’ regions, discovered through whole chloroplast genome analysis, as potential barcode markers for species within the genus Pyropia. In the end the authors settle on an intergenic region, trnM-argB as the most appropriate marker due to a combination of marker length, primer universality and resolving power.

The manuscript is concise and well written and the information presented could be of value to researchers requiring easy molecular identification of Pyropia species and potentially population level markers. However, I believe that the authors need to clarify a couple of points concerning the usefulness of these new markers before publication. These concerns are detailed below.

Although I am not going to argue the need for good species level markers in many algal groups, I believe this would be a stronger manuscript if the authors more thoroughly explained the need for new molecular markers within the genus Pyropia. They carefully describe the limitations of morphological species delimitation and their previous PCR-RFLP method. However, several studies (e.g. Milstein et al. 2015 and Yang et al. 2018) have successfully used 18S and rbcL for species level phylogenies within Pyropia. Yang et al. in particular found good species level resolution using rbcL and standard species delimitation methods. Please clarify how this study improves upon what is currently available to researchers looking for molecular identification of Pyropia species.

Given the title of manuscript I was interested in whether the authors had used this comparative genomic method to identify hypervariable markers that could resolve population-level associations within the genus, as this is largely a shortcoming of 18S and rbcL. In fact the authors state in the last sentence of the manuscript that these markers can now be employed in population and phylogenetic studies. While this would be of significant interest, by only including six distinct species in this study the authors present no evidence that these markers have the resolving power to distinguish sub-species relationships or would be useful in phylogenetic analyses (there may be alignment issues with markers with so many insertions and deletions). I think that a genetic distance comparison with rbcL, the inclusion of multiple samples from within a species, or some additional evidence may be required if statements concerning their population-level usefulness are going to be included.

Finally, in the last paragraph of the results the authors state that the trnM-argB marker discriminated between the six species with an average pairwise distance of 0.295. It may be more useful to present the entire genetic distance matrix for the six species as opposed to the average distance. This would give the reader a better appreciation for the power of this marker between closely related species.

Author Response

Response to Reviewer#2:

We, the authors of diversity-639877, would like to thank the reviewer for careful and thorough reading of this manuscript and for the thoughtful comments and constructive suggestions, which help to improve the quality of this manuscript. We believe that the comments and the clarity of our paper are much improved in the revised version. Our response follows (the reviewer's comments are in italics).

General Comments.

1) However, several studies (e.g. Milstein et al. 2015 and Yang et al. 2018) have successfully used 18S and rbcL for species level phylogenies within Pyropia. Yang et al. in particular found good species level resolution using rbcL and standard species delimitation methods. Please clarify how this study improves upon what is currently available to researchers looking for molecular identification of Pyropia species.

Reply:

However, the resolution of these markers may not be optimal between P. dentata and P. haitanensis, particularly at the genus level, with rbcL known to provide lower resolution than 18S in some cases. Therefore, advances in NGS technologies have made it possible to sequence whole cp genomes and identify molecular markers. Highly variable markers derived from the cp genomes of different species at the genus level have uncovered many loci that are informative for systematic marine taxa and aquaculture industry.

2) I think that a genetic distance comparison with rbcL, the inclusion of multiple samples from within a species, or some additional evidence may be required if statements concerning their population-level usefulness are going to be included.

Reply:

Discussion section exist to answer the reviewer's suggestion;

Although the present study analyzed a limited number of Pyropia species and chloroplast genomes, further studies should include various other species and chloroplast genomes to further elucidate the phylogentic relationships of the Pyropia genus.

3) Finally, in the last paragraph of the results the authors state that the trnM-argB marker discriminated between the six species with an average pairwise distance of 0.295. It may be more useful to present the entire genetic distance matrix for the six species as opposed to the average distance.

Reply:

The suggested correction has been made.

P. yezoensis

P. dentata

P. suborbiculata

P. haitanensis

P. seriata

P. pseudolinearis

P. yezoensis

P. dentata

0.295

P. suborbiculata

0.28

0.291

P. haitanensis

0.307

0.123

0.291

P. seriata

0.33

0.215

0.349

0.234

P. pseudolinearis

0.295

0.261

0.272

0.234

0.28

Reviewer 3 Report

Overall, I find this a very clear and direct manuscript.  It takes advantage of available chloroplast genome (plastome) data to identify potential genetic markers for a group of ecological and economically important seaweeds.  The main result, that there are indels in the trnM-argB region which can serve as species markers, seem reasonably well supported within limits that are, unfortunately, not explored or discussed by the authors.

There is no mention of the level of intrapopulation or intraspecific variation in these markers. 

I know this to be an issue in some microalgae.  My lab is utilizing the plastome to map population level phylogeny in several groups of microalgae.  These populations are lacustrine and so are relatively isolated from one another.  We intrapopulation (intralacustrine assemblage) variation in the plastome. 

It appears that the authors only sample one individual per species. I apologize if I missed a description of this (if I did, the authors might want to make explicit through standard notation in Table 2 (e.g., after the species name, N=3, or some such.)

One might be tempted to consider this not a problem if one understood more about the biology of these algae, and/or if the authors addressed the issue in general terms.

The biology of these species is not described to the reader.  Intraspecific variation has several sources according to the literature.  One is simply that there are variation in markers in a population.  Another is heteroplasmy (variation within an individual).  The latter can result from either retention of variants among chloroplasts in a lineage, and can be exacerbated by complicated (e.g., biparental) inheritance of plastids. 

None of this may be a problem.  However, I (and most readers) are not familiar enough with Pyropia to know how plastids are inherited.  If there is good evidence for uniparental inheritance, then heteroplasmy is less likely to be an issue.

In short, I think this paper would benefit from a discussion of the possibility of intraspecific variation and biology of these algae.

Another limitation is that this paper could put the diversity studied in better perspective for the reader.  I have seen quite a few papers from Korean labs that focus solely on species in Korea, with little to no mention of the distribution of species studied beyond Korea.  How far up the Russian coast do these species occur (if at all?)  What is their distribution in the Yellow Sea or Japan?  Perhaps more importantly, are there closely related species not mentioned that occur in nearby locations, but NOT in Korea.  I.e., I do not have a good sense of just how broadly applicable these results will be.

Admittedly, the authors recognize the need to sample other species (in the last paragraph) but I think this could and should be better developed.

I don't think any of these are fatal flaws, but without a discussion of these factors and issues, this manuscript is simply a technical report (we sampled 6 things and found genetic differences) with little biology or other information that might be useful to a broader audience interested in marine diversity. 

If the authors and editor are comfortable with that (and it may well be appropriate to the journal), then perhaps all these issues need not be addressed, except for intraspecific variation, which is still an issue of critical importance.

Author Response

Response to Reviewer#3:

We, the authors of diversity-639877, would like to thank the reviewer for careful and thorough reading of this manuscript and for the thoughtful comments and constructive suggestions, which help to improve the quality of this manuscript. We believe that the comments and the clarity of our paper are much improved in the revised version. Our response follows (the reviewer's comments are in italics).

General Comments.

1)It appears that the authors only sample one individual per species. I apologize if I missed a description of this (if I did, the authors might want to make explicit through standard notation in Table 2 (e.g., after the species name, N=3, or some such.)

Reply:

we tested three individuals in six Pyropia species. The suggested correction has been made.

2)Another limitation is that this paper could put the diversity studied in better perspective for the reader. I have seen quite a few papers from Korean labs that focus solely on species in Korea, with little to no mention of the distribution of species studied beyond Korea.  How far up the Russian coast do these species occur (if at all?)  What is their distribution in the Yellow Sea or Japan?  Perhaps more importantly, are there closely related species not mentioned that occur in nearby locations, but NOT in Korea.  I.e., I do not have a good sense of just how broadly applicable these results will be.

Reply:

Discussion section exist to answer the reviewer's suggestion;

Although the present study analyzed a limited number of Pyropia species and chloroplast genomes, further studies should include various other species and chloroplast genomes to further elucidate the phylogentic relationships of the Pyropia genus.